# CLoSD: Closing the Loop between Simulation and Diffusion for Multi-task Character Control

**Guy Tevet**[1,2]**, Sigal Raab**[1]**, Setareh Cohan**[2]**, Daniele Reda**[2,3]**, Zhengyi Luo**[4]**,
Xue Bin Peng**[5,6]**, Amit H. Bermano**[1]**, and Michiel van de Panne**[2]

[1]Tel-Aviv University, [2]University of British Columbia, [3] Wayve,
[4]Carnegie Mellon University, [5]Simon Fraser University, [6]NVIDIA
guytevet@mail.tau.ac.il

## Abstract

Motion diffusion models and Reinforcement Learning (RL) based control for physics-based simulations have complementary strengths for human motion generation. The former is capable of generating a wide variety of motions, adhering to intuitive control such as text, while the latter offers physically plausible motion and direct interaction with the environment. In this work, we present a method that combines their respective strengths. CLoSD is a text-driven RL physics-based controller, guided by diffusion generation for various tasks. Our key insight is that motion diffusion can serve as an on-the-fly universal planner for a robust RL controller. To this end, CLoSD maintains a closed-loop interaction between two modules — a Diffusion Planner (DiP), and a tracking controller. DiP is a fast-responding autoregressive diffusion model, controlled by textual prompts and target locations, and the controller is a simple and robust motion imitator that continuously receives motion plans from DiP and provides feedback from the environment. CLoSD is capable of seamlessly performing a sequence of different tasks, including navigation to a goal location, striking an object with a hand or foot as specified in a text prompt, sitting down, and getting up.

https://guytevet.github.io/CLoSD-page/

## 1 Introduction

Physics-based character animation leverages simulations to generate plausible human movement, particularly in terms of realistic contacts, and interactions with objects and other characters in a scene. To achieve a set of given goals or a task, a control policy is often synthesized using Reinforcement Learning (RL). As shown in recent years (Peng et al., 2018; 2021), RL greatly benefits from leaning on kinematic motion capture data as a reference in order to learn to produce human-like behaviors. Such methods typically rely on individual clips or small-scale curated datasets to train controllers for individual tasks, as this simplifies the RL exploration problem and allows for compact models of the underlying desired motion manifold. However, it is often difficult to scale RL methods to leverage larger human motion datasets, such as those commonly used to train kinematic motion generation models (Tevet et al., 2023; Dabral et al., 2023).

In this paper, we present *Closing the Loop between Simulation and Diffusion*, dubbed *CLoSD*, a method that allows for closed-loop operation of text-to-motion diffusion models with physics-based simulations. Our key insight is that motion diffusion, given textual instruction and a target location, can serve as a versatile kinematic motion planner. A physics-based motion tracking controller then serves as the *executor* of the plan in simulation and facilitates physically plausible interactions with the environment. However, motion diffusion models are typically used for offline motion generation, with slow inference times due to the gradual denoising process. Instead, we require a model that reacts in real-time and whose movements are grounded in the scene. To this end, we introduce Diffusion Planner (DiP), an auto-regressive real-time diffusion model, generating high-fidelity motions using as few as 10 diffusion steps, while being conditioned on both text and target location for

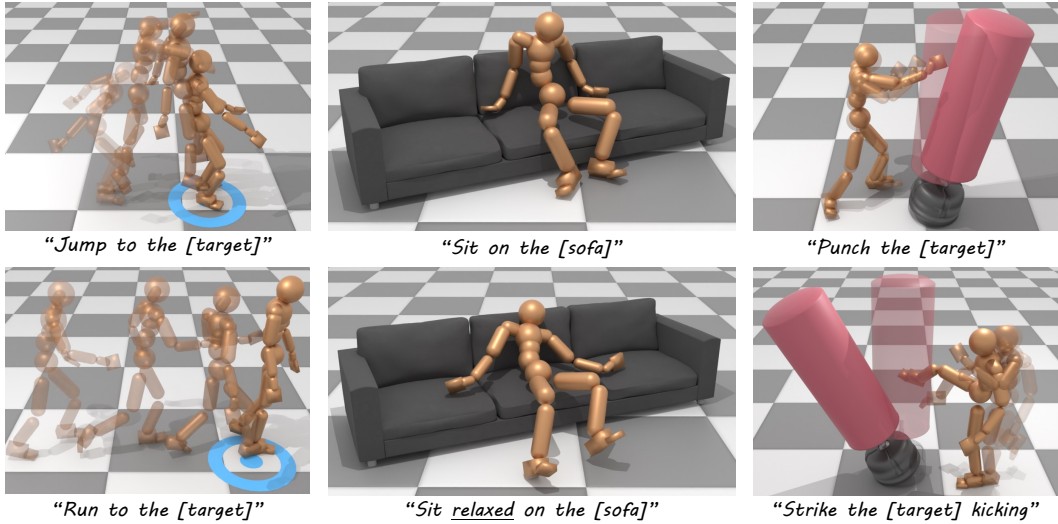

"Jump to the [target]"    "Sit on the [sofa]"    "Punch the [target]"

"Run to the [target]"    "Sit _relaxed_ on the [sofa]"    "Strike the [target] kicking"

Figure 1: **CLoSD** is a multi-task physics-based RL controller, capable of performing object interaction tasks and can be controlled through a text prompt and target positions. The engine of CLoSD is the Diffusion Planner (DiP), a real-time diffusion model that operates in a closed feedback loop with the RL, producing high-fidelity human motions for completing a diverse set of tasks.

arbitrary joints (e.g. "Strike with a hook punch" + *left hand target* or "perform a high-kick" + *right foot target*).

We use a robust RL tracking controller to execute the DiP plan. As demonstrated in PHC (Luo et al., 2023), a general motion tracking policy can enable physically simulated characters to follow a wide spectrum of human motions. The use of a physics simulation can also correct for nonphysical artifacts in the input target motion, such as floating, sliding, and penetration with objects. We synergize our diffusion-based planner and the motion tracker by forming a closed feedback loop between them where CLoSD enables simulated characters to produce robust and responsive environment-aware behaviors. The executed motion of the simulated character via the physics simulation is fed back to the diffusion model auto-regressively and becomes the prefix of the next sequence. Then, given the text prompt and a task-related goal, e.g., the target location to strike, DiP generates the next motion plan for the policy, closing the planning-execution loop. Unlike typical offline motion diffusion, the closed feedback loop in CLoSD enables interactive text-driven control, where users can change text commands on the fly and our system is able to seamlessly transition between the desired behaviors.

Nonetheless, a mismatch occurs as the original motion tracker was trained neither considering object interaction, nor interactions with a real-time planner. We therefore fine-tune the tracking RL policy with DiP in-the-loop. This process is universal and is performed for multiple tasks at once. The fine-tuned tracker is then robust to the closed-loop state sequences as observed for the object interactions.

We show that a single instance of CLoSD successfully performs goal-reaching, target-striking, and couch interactions, each of which required a dedicated policy and considerable reward engineering in prior work (Peng et al., 2022; Hassan et al., 2023). The results of our experiments demonstrate that CLoSD outperforms both the current leading text-to-motion controller and the state-of-the-art multi-task controller. Our diffusion planner generates 40-frame plans at 3,500 fps, i.e., $175\times$ real-time, allowing ample scope for real-time use even with future additions.

## 2    RELATED WORK

Recent advances in kinematic and physics-based methods for human motion generation enable a variety of text-to-motion, motion imitation, and other task specification methods.

**Data-driven motion synthesis.** Advances in language and generative models have led to the emergence of flexible and intuitive text-to-motion models for computer animation. The language and semantic understanding encapsulated in text encoders (Radford et al., 2021; Devlin, 2018), together with the multi-modal generative capabilities of diffusion models (Sohl-Dickstein et al., 2015; Song & Ermon, 2020) have given rise to a new generation of high-fidelity generative models, particularly for images and videos (Rombach et al., 2022; Ho et al., 2022). This approach has been successfully adapted to the motion synthesis domain (Tevet et al., 2022; 2023; Dabral et al., 2023; Zhu et al., 2023), providing powerful tools for a large variety of motion synthesis and editing tasks (Shafir et al., 2024; Tseng et al., 2023; Cohan et al., 2024; Goel et al., 2024; Karunratanakul et al., 2024). Recently, MoMo (Raab et al., 2024a), MAS (Kapon et al., 2024) and SinMDM (Raab et al., 2024b) show generalization to out-of-domain motions using various diffusion methods. A-MDM (Shi et al., 2024) and CAMDM (Chen et al., 2024) presented auto-regressive motion diffusion models and demonstrated interactive control tasks. Xie et al. (2023) used a diffusion model learned on limited data as a reference for a box loco-manipulation task. Trace and Pace (Rempe et al., 2023) used a diffusion model to plan character trajectories on a 2D map for character navigation.

**Diffusion models for object interaction.** Works in both Human-Object Interaction (HOI) and Human-Scene Interaction (HSI) present interaction with objects, e.g., sitting on a couch. However, HOI focuses on interaction with individual objects, while HSI addresses interaction with the entire scene. While CLoSD offers both textural control and a physics-based control, current diffusion-based HOI and HSI works typically allow either textual input (Wu et al., 2024; Peng et al., 2023; Wang et al., 2024; Yi et al., 2024; Cen et al., 2024; Li & Dai, 2024) or physical-based methods (Huang et al., 2023), or neither Kulkarni et al. (2024).

**Text controlled Simulated Characters.** While data-driven motion synthesis models are able to generate life-like motions, they are still prone to exhibiting artifacts such as foot sliding and floating. Further, creating realistic human-object interactions using purely data-driven approaches is challenging, as the underlying contact dynamics and collisions are difficult to learn directly from data. Using physics-simulated characters (Juravsky et al., 2022; 2024; Yao et al., 2024; Truong et al., 2024; Xiao et al., 2024; Ren et al., 2023) provides a solution by enforcing physically plausible constraints on motion and interaction. Among them, the PADL line of work uses natural language to control simulated characters, starting from simple instructions (Juravsky et al., 2022) to scale up to more complex instructions (Juravsky et al., 2024). MoConVQ (Yao et al., 2024) utilizes an LLM to direct characters using a pretrained motion latent space and in-context learning, but neither approach supports fine-grained human-object interactions. UniHSI (Xiao et al., 2024) uses LLMs to specify target positions for controlling a simulated humanoid in human-object interaction tasks. However, as only the target end-effector position is used as the control interface, the style of the motion cannot be influenced by language. PDP (Truong et al., 2024) learns from an offline humanoid motion imitation dataset utilizing a diffusion policy (Chi et al., 2023) and demonstrates an early attempt at learning language-based control from offline data. InsActor (Ren et al., 2023) utilizes a differentiable physics simulator and diffusion model to achieve text-guided humanoid control. Compared to InsActor, we enable human-object interactions and alleviate artifacts introduced by the differentiable simulator. In contrast, we propose to bridge the gap between data-driven kinematic text-to-motion generation and physics-based character control using a closed-loop plan-and-imitate system.

**Humanoid Motion Imitation.** Simulated humanoid motion imitation has seen significant advancements in recent years (Al Borno et al., 2012; Peng et al., 2018; 2019; Chentanez et al., 2018; Won et al., 2020; Yuan & Kitani, 2020; Fussell et al., 2021; Luo et al., 2023; 2021; 2022; Wang et al., 2020). Since no ground-truth action annotations exist and physics simulators are often non-differentiable, policies, imitators, or controllers are trained to track, imitate, or mimic human motion using deep reinforcement learning (RL). From policies that can track a single clip of motion Peng et al. (2018), to large-scale datasets Won et al. (2020); Luo et al. (2023), motion imitators become more and more useful as they can reproduce an ever-growing corpus of behaviors. In this work, we utilize PHC Luo et al. (2023), an off-the-shelf motion imitator that has successfully scaled to the AMASS dataset Mahmood et al. (2019), as the foundation for our approach due to its robustness in handling diverse kinematic motions. Recent works leveraged imitation policies to learn multi-purpose controllers; PULSE (Luo et al., 2024) distilled it to a latent action space for down-stream tasks, whereas MaskedMimic(Tessler et al., 2024) learned a multi-modal controller based on partial information imitation.

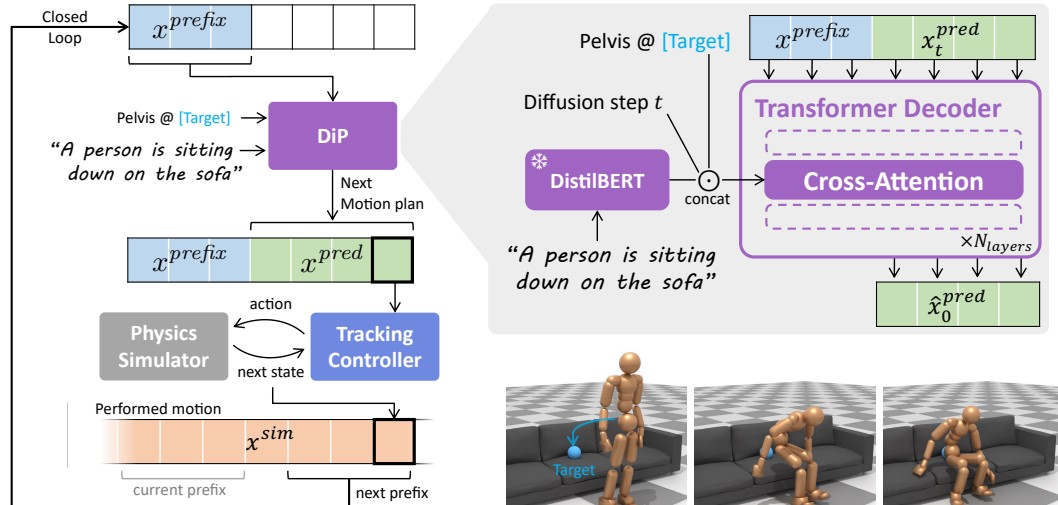

Figure 2: **CLoSD Overview. (Left)** DiP is a rapid auto-regressive diffusion model conditioned on a text prompt and a Target location. It generates the motion plan $x^{\text{pred}}$ which is then performed by the RL Tracking Controller. The result frames are fed back to DiP as $x^{\text{prefix}}$. Together they maintain planning-execution loop. **(Top right)** A zoom-in to a single DiP denoising iteration.

## 3   PRELIMINARIES

**Motion representation.** We use two different motion representations, each suitable for its purpose. For the *kinematic motions* generated by the diffusion model, $x$, we follow MDM (Tevet et al., 2023) and use the HumanML3D motion representation, in which each frame is represented *relatively* to the previous one. For the RL tracking policy, we follow PHC (Luo et al., 2023) and represent the motion, $x^{\text{sim}}$ (i.e. a sequence of states) by their *global* position and velocity, at the world axis. To convert from HumanML3D to PHC we implement $x^{\text{sim}} = R2G(x)$, (stands for relative-to-global) using simple accumulation. To convert from PHC to HumanML3D we use $x = G2R(x^{\text{sim}})$, (stands for global-to-relative) as implemented for HumanML3D. The rotations are calculated using first-order inverse kinematics and the foot contacts using a height heuristic. We simulate a humanoid robot compatible with SMPL (Loper et al., 2015) joints, hence, converting the simulated results back to the SMPL mesh is straightforward (Figure 3).

**Denoising diffusion models.** The DDPM (Ho et al., 2020) framework used in this paper models diffusion as a Markov noising process, $\{x_t\}_{t=0}^T$, where $x_0$ is drawn from the data distribution and $q(x_t|x_{t-1}) = \mathcal{N}(\sqrt{\alpha_t}x_{t-1}, (1 - \alpha_t)I)$, where $\alpha_t \in (0, 1)$ are constant hyper-parameters. When $\alpha_t$ is small enough, we can approximate $x_T \sim \mathcal{N}(0, I)$. In our context, conditioned motion synthesis models the distribution $p(x_0|c)$ as the reversed diffusion process of gradually cleaning $x_T$. Instead of predicting $\epsilon_t$ as formulated by Ho et al. (2020), we follow MDM (Tevet et al., 2023) and use an equivalent formulation to predict the motion sequence itself, i.e., $\hat{x}_0$ with the *simple* objective (Ho et al., 2020), $\mathcal{L}_{\text{simple}} = E_{x_0 \sim p(x_0|c), t \sim [1,T]}[\|x_0 - \hat{x}_0\|_2^2]$. Predicting $\hat{x}_0$ is found to produce better results for motion data, and enables us to apply a target loss as a geometric loss at each denoising step.

## 4   CLoSD

CLoSD is a text-driven character controller capable of performing multiple tasks involving physically-based object interactions. An overview is given in Figure 2. The input to CLoSD consists of (a) a text prompt describing the desired motion, including its style, content, etc., and (b) a target location for one joint of the character. The combination of these two inputs provides a simple but expressive description for object interaction tasks. For example, the text prompt "strike the target using a high kick" combined with the target location for the *right foot* at the target location, precisely defines how to strike it (Figure 1), while the prompt "strike the target by pushing it" combined with

the same target location, this time for one of the *hand joints*, defines a completely different strike. Note that both inputs can be changed online to enable interactive control by the user and can be sequenced to perform a series of different tasks.

CLoSD has two key components, Diffusion Planner (DiP, §4.1), a real-time diffusion model that synthesizes kinematic motion plan, and a robust motion tracking controller (§4.2) that then executes the desired motion. These then operate in a closed-loop fashion (§4.3) by providing the realized motion as an input prefix to DiP, which enables DiP to react to the physical environment. A simple state machine (§4.4) detects task completions and transitions to the next task by changing the text prompt and the target on the fly. This enables the smooth completion of a sequence of multiple tasks (see the web page video).

## 4.1 MOTION DIFFUSION PLANNER

To guide CLoSD to perform multiple tasks in a physically simulated environment, we use a kinematic motion generator as a planner. Motion diffusion models such as MDM (Tevet et al., 2023), are a leading method for motion synthesis but are not suitable for several reasons. They are designed to generate end-to-end motions in an offline manner and are therefore slow, making them incompatible with real-time interaction with a physics-based simulation. Furthermore, the large text-to-motion benchmarks on which MDM was trained, lack any description or awareness of the environment.

To overcome these limitations, we designed a diffusion planner (DiP), a real-time diffusion model that is aware of both the current state of the agent and the task to be performed. State awareness is achieved by autoregressively updating the motion plan, i.e. the diffusion planner receives an input motion prefix representing the previous motion frames and outputs a prediction of the frames ahead. The model is further conditioned on (1) a text prompt specifying the task (e.g. "A person is sitting on the sofa") and (2) target locations for one of the body joints (e.g. pelvis joint at the location of the sit). The latter serves to ground the task description to specific physical locations.

DiP follows the principles presented by Chen et al. (2024) in the design of a real-time auto-regressive diffusion model with a transformer decoder backbone. Figure 2 (top right) illustrates a single denoising diffusion step. At each step $t \in [0, T]$, the model gets as input the clean prefix motion $x^{\text{prefix}}$ to be completed, the noisy motion prediction $x_t^{\text{pred}}$ to be denoised, the diffusion step $t$, and the two conditions - a text prompt, and a target location (optionally more than one). Here, $x^{\text{prefix}}$ and $x_t^{\text{pred}}$, with fixed lengths $N_p$ and $N_g$, respectively, are concatenated, added to standard positional embedding tokens to indicate their indices, and input to the transformer decoder backbone, such that each frame is represented by a single input token. From the transformer's output, the last $N_g$ frames corresponding to $x_t^{\text{pred}}$ are the clean motion prediction $\hat{x}_0^{\text{pred}}$ whereas the first $N_p$ output frames corresponding to the prefix are not in use.

Simultaneously, the model's conditions are constructed and injected through the cross-attention blocks of each of the transformer's layers. The text condition is encoded using a fixed instance of DistilBERT (Sanh, 2019) to a sequence of latent tokens, followed by a learned linear layer to coordinate the dimensions, resulting in $C_{\text{text}} \in \mathbb{R}^{N_{\text{tokens}} \times d}$, where $d$ is the transformer's dimension. DistilBERT is a lighter version of the popular text encoder BERT (Devlin, 2018), which is not biased towards a global description of images, as CLIP (Radford et al., 2021), and as a result, was widely used for text-to-motion (Petrovich et al., 2022; Athanasiou et al., 2022).

The diffusion step $t$ and the target condition are embedded through two separate shallow fully connected networks to $C_t$ and $C_{\text{target}}$ correspondingly, added together and concatenated to $C_{\text{text}}$ to conform the full condition sequence $(C_t + C_{\text{target}}, C_{\text{text}}) \in \mathbb{R}^{(N_{\text{tokens}}+1) \times d}$. This condition is injected into each of the transformer decoder's layers through their cross-attention blocks.

**Adaptive target conditioning.** The target condition $c_j \in \mathbb{R}^3$ specifies the desired location of joint $j \in J$ at the last predicted frame, where J is the subset of end-effector joints and the pelvis. Since different tasks require different joints, we want to adaptively condition DiP on different subsets of joints, and be able to change them on the fly as the task changes. To this end, we add a boolean validity signal $v_j$ per joint which indicates if it is currently used as a condition. In addition, we condition DiP on the heading angle of the body in the $XY$ plane, $c_\theta \in \mathbb{R}$, and its validity signal $v_\theta$ respectively. The full target condition is hence a concatenation of all joint locations and heading angle $(\{c_j\}_{j \in J}, \{v_j\}_{j \in J}, c_\theta, v_\theta)$.

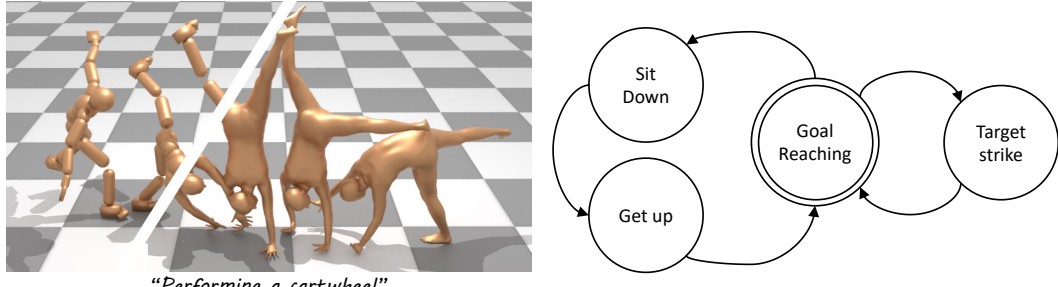

"Performing a cartwheel"

Figure 3: **(Left)** CLoSD generates versatile text-prompted physics-based motions. The SMPL-compatible physics model is rendered with the SMPL mesh. **(Right)** CLoSD can perform a sequence of RL tasks (see the web page video). Task transitions are user-specified via interactively changing the text, or via a state machine, with transitions on a task *done* signal.

**Target loss.** Reaching the target condition at the last predicted frame $\hat{x}_0[N_g]$ is enacted via an additional loss term $\mathcal{L}_{\text{target}}$. Since DiP predicts the motion $\hat{x}_0$ at each diffusion step $t$, rather than the noise $\epsilon_t$, we can follow MDM and implement the target loss as a geometric loss as follows:

$$\mathcal{L}_{\text{target}} = \sum_{j \in J} v_j ||R2G(\hat{x}_0)_j[N_g] - c_j||_2^2 + v_\theta ||R2G(\hat{x}_0)_\theta[N_g] \ominus c_\theta||_2^2$$

where $\ominus$ denotes angles difference and $R2G$ is a differentiable relative-to-global representation converter. The last is needed since HumanML3D representation (See Section 3) defines each frame *relative* to the previous frame whereas $C_{\text{target}}$ defines the target in a *global* coordinates system.

**Training.** During training we sample (motion, text) pairs from the dataset, and a denoising timestep $t \sim \mathcal{U}[0, T]$. The motion is cropped to a length $N_p + N_g$ with a random offset, then split to $x^{\text{prefix}}$ and $x_0^{\text{pred}}$ respectively. $x_0^{\text{pred}}$ is noised to $t$ resulting $x_t^{\text{pred}}$. The text prompt is encoded to $C_{\text{text}}$ using the frozen DistilBERT and $t$ to $C_t$ using the dedicated shallow network. To learn the random target condition, we take the heading angle and a joint position of a randomly sampled joint $j \in J$ from the last frame of $x^{\text{pred}}$, encode them to the input $C_{\text{target}}$ and use them as the ground truth $c_j$ and $c_\theta$ for the target loss $\mathcal{L}_{\text{target}}$. DiP gets the above and predicts $\hat{x}_0^{\text{pred}}$. The full loss applied to the model is then $\mathcal{L} = \mathcal{L}_{\text{simple}} + \lambda_{\text{target}} \mathcal{L}_{\text{target}}$.

**Inference**. We follow the standard DDPM iterative inference for $x_0$ prediction model: Initializing $x_T^{pred} \sim \mathcal{N}(0, I)$, then for each $t \in [T, 1]$ we let DiP to predict $\hat{x}_0^{pred}$, partially noise it to $x_{t-1}^{pred}$ and repeat. At the last iteration, we use DiP's output as the final prediction $\hat{x}^{pred}$.

Unlike MDM, which is known for its high runtime cost, DiP is substantially faster. It generates 2 seconds of reference motion in 11.4 ms, i.e., $175\times$ faster than real time. This acceleration is achieved by predicting shorter motion segments and reducing the number of diffusion steps. Predicting shorter segments helps since the self-attention runtime depends quadratically on the sequence length. Additionally, as shown by Chen et al. (2024), auto-regressive MDM requires fewer diffusion steps; we are able to reduce the number of diffusion-denoising steps from 50 in MDM to 10, and accelerate generation by a factor of 5.

## 4.2 UNIVERSAL TRACKING POLICY

To perform DiP's planning in a physically simulated environment, we leverage a motion tracking policy that translates the kinematic motions to control forces that can be used to drive the movement of a simulated humanoid character. To this end, we use a slim version of the Perpetual Humanoid Control (PHC) (Luo et al., 2023), i.e. a single PHC policy without the suggested mixture of experts. PHC is a universal RL tracking policy, capable of imitating the large AMASS motion capture dataset with 99% success rate, where successful imitation is assumed when the maximal per-joint error is less than $0.5[m]$, for all joints during the entire motion. PHC implements a Markov Decision Process (MDP) policy $\pi_{\text{PHC}}$ that at each simulation step $n$ observes the current humanoid pose $x^{\text{sim}}[n]$ and the current pose to be imitated $x^{\text{ref}}[n]$ and outputs the humanoid action $a[n]$ driving it to perform $x^{\text{ref}}[n]$ at the next step. In detail, we use PHC's keypoint-only state representation

$s[n] = (x^{\text{ref}}[n] - x^{\text{sim}}[n], x^{\text{ref}}[n])$. PHC uses a proportional derivative (PD) controller at each DoF of the humanoid such that the action $a[n]$ specifies the PD target. The goal reward is the negative exponent of the distance between $x^{\text{ref}}[n]$ and $x^{\text{sim}}[n+1]$ accompanied by adversarial reward as defined in AMP (Peng et al., 2021) and energy penalty. PHC is trained using the Proximal policy optimization (Schulman et al., 2017) (PPO). For full details of the implementation, we refer the reader to (Luo et al., 2023).

While PHC uses progressive neural networks with multiple primitives (policies) to achieve robust fall recovery, training a single primitive for a longer time yields a comparable imitation success rate. Since we desire a lightweight tracking policy that can be easily finetuned for multi-task training and we do not require fall recovery, we use the single primitive $\pi_{\text{PHC}}$.

### 4.3 CLOSING THE LOOP

As shown in Figure 2, the reference motion provided as input to $\pi_{\text{PHC}}$ is the DiP predicted plan, converted to the PHC representation, $x^{\text{ref}} = R2G(x^{\text{pred}})$. Whereas DiP generates $N_g$ long motion segments at a time, $\pi_{\text{PHC}}$ processes them frame-by-frame: for each frame $n$, $\pi_{\text{PHC}}$ observes $x^{\text{ref}}[n]$ and $x^{\text{sim}}[n]$ and outputs $a[n]$. The simulator receives the action and outputs the next state of the humanoid which is buffered as $x^{sim}[n]$. When the policy-simulator iterations reach the end of $x^{\text{ref}}$, The last $N_p$ frames of $x^{\text{sim}}$ are cropped and used as the new input $x^{\text{prefix}} = G2R(x^{\text{sim}}[-N_p:])$ to DiP, after converting back to the relative representation, using the $G2R$ block. DiP in turn generates a new $x^{\text{pred}}$ to be consumed by $\pi_{\text{PHC}}$.

**Fine-tuning.** Despite PHC's impressive motion tracking capabilities, there is a gap between the motion tracking task $\pi_{\text{PHC}}$ was trained on and the capabilities required by CLoSD. First, $\pi_{\text{PHC}}$ was trained using a relatively realistic motion capture dataset. While motion diffusion models are able to generate high-fidelity motions, they are still prone to generating physically inaccurate motions (Yuan et al., 2023). It means that $\pi_{\text{PHC}}$ will need to fix such artifacts on the fly and avoid motions that might cause falling. Second, $\pi_{\text{PHC}}$ was trained to track motion capture without object interaction, and therefore does not learn the control required for interaction.

For those reasons, we fine-tune $\pi_{\text{PHC}}$ in a closed loop manner using Proximal Policy Optimization (PPO) (Schulman et al., 2017). Since both CLoSD and the imitation $\pi_{\text{PHC}}$ are universal components, and since neither reward nor reset engineering is needed, we can fine-tune the same policy for multiple tasks simultaneously. In practice, during fine-tuning, we fix DiP and for each episode randomly choose a task, then set the object, text prompt, and target location accordingly. We use the original PHC rewards and reset conditions, i.e., it is still trained to track a target motion, where the target motion is specified by DiP and involves different types of object interactions. The fast inference time of DiP and its relatively low memory usage enable training without compromising the large number of parallel environments required for PPO.

### 4.4 HIGH-LEVEL PLANNING USING A STATE MACHINE

CLoSD is a multi-task agent. Each task is defined by the text prompt and the target location conditions. Since those conditions can be changed on the fly by the user, CLoSD is capable of performing a sequence of different tasks at test time by simply transitioning between different target conditions. We automate the transitions using a simple *state machine* to demonstrate this. An example state-machine is illustrated in Figure 3. To choose a meaningful transition point between tasks, each task signals when it is done. For example, *sitting* is done when the pelvis is on the sitting area of the sofa and *striking* is done when the target hits the ground. When a task is done, the state machine can randomly select a new one. Because the planning burden is taken care of by DiP, high-level planning for CLoSD can generally be quite simple.

## 5 EXPERIMENTS

We measure the performance of CLoSD using two experiments. *Task performance* (§5.2) measures its success rate in goal reaching, target striking, sitting, and getting up. *Text-to-motion* (§5.3) measures its ability to apply to text prompts and span the motion manifold according to the popular

HumanML3D (Guo et al., 2022) benchmark and compares it to purely data-driven models. The code is available at `https://github.com/GuyTevet/CLoSD`.

## 5.1 IMPLEMENTATION DETAILS

**Motion representations.** We use two different motion representations, each suitable for its purpose. For the *kinematic motions* generated by DiP, we follow MDM (Tevet et al., 2023) and use the HumanML3D motion representation. Each frame $n$ (i.e. a single pose) of the motion sequence $x$ is defined by $x[n] = (\dot{r}^a, \dot{r}^x, \dot{r}^z, r^y, j^p, j^r, j^v, f) \in \mathbb{R}^F$, where $\dot{r}^a \in \mathbb{R}$ is the root angular velocity along the Z-axis. $\dot{r}^x, \dot{r}^z \in \mathbb{R}$ are root linear velocities on the XY-plane, and $r^y \in \mathbb{R}$ is the root height. $j^p \in \mathbb{R}^{3(J-1)}$, $j^r \in \mathbb{R}^{6(J-1)}$ and $j^v \in \mathbb{R}^{3J}$ are the local joint positions, velocities, and rotations with respect to the root, and $f \in \mathbb{R}^4$ are binary features denoting the foot contact labels for four-foot joints (two for each leg). For the RL tracking policy, we follow PHC (Luo et al., 2023) and represent a frame (i.e. a single state) as $x^{\text{sim}}[n] = (j^{gp}, j^{gv})$ where $j^{gp} \in \mathbb{R}^{3J}$ are the joint's global positions, and $j^{gv} \in \mathbb{R}^{3J}$ are the joint's linear velocities relative to the world (as opposed to $j^v$ which is relative to the root).

**DiP** is implemented as an 8 layer transformer decoder with a latent dimension of 512 and 4 attention heads. The text encoding is done using a fixed instance of DistilBERT (Sanh, 2019). It was trained on the HumanML3D text-to-motion dataset using the MDM (Tevet et al., 2023) framework for 600K diffusion steps on a single *NVIDIA GeForce RTX 3090* GPU. We find the 200K checkpoint optimal for CLoSD and use it across our experiments. We use 10 diffusion steps, prefix length $N_p = 20$ and generation length $N_g = 40$ as we find it a good trade-off between quality and runtime (Table 2). This instance of DiP impressively outputs 3500 frames per second on the same GPU. The classifier-free guidance parameter used for DiP-only, the task experiments and the text-to-motion experiment are 7.5, 2.5, and 5 respectively.

**Tracking policy.** We follow PHC (Luo et al., 2023) and train the joints only (without rotation angles tracking) single-primitive tracking policy over the AMASS (Mahmood et al., 2019) dataset for 62K PPO epochs. We further fine-tune it in the closed loop as described in Section 4.3 for 4K more epochs. We train using the Isaac Gym simulator (Makoviychuk et al., 2021) with 3072 parallel environments on a single *NVIDIA A100* GPU.

**Baselines.** To understand the merits of the closed loop design, we suggest the open loop baseline, in which DiP generates the motion offline, fed by its own prediction, and then the tracker follows this fixed trajectory. In addition, we report the performance of CLoSD before fine-tuning and experimenting with longer ($N_g = 80$) and shorter ($N_g = 10$) plans, as ours is 40 frames.

## 5.2 TASK EVALUATION

The following are the descriptions of the tasks to be evaluated. We fine-tune the tracking controller (closed loop) for all tasks simultaneously, without any reward engineering. Each environment is randomly initialized to perform one of the tasks. DiP performs planning according to the relevant text prompt and target location and the policy is fine-tuned with PPO with the original PHC objectives.

| | Goal reaching | Object striking | Sitting | Getting up |
|---|---|---|---|---|
| AMP (2021) reach | 0.88 | - | - | - |
| AMP (2021) strike | - | 1.0 | - | - |
| InterPhys (2023) sit | - | - | 0.76 | - |
| UniHSI (2024) | 0.96 | 0.02 | 0.85 | 0.08 |
| CLoSD (Ours) | **1.0** | **0.9** | **0.88** | **0.98** |
| shorter-loop | 0.86 | 0.71 | 0.61 | 0.95 |
| longer-loop | 0.99 | 0.86 | 0.56 | 0.92 |
| w.o. fine-tuning | **1.0** | 0.81 | 0.66 | 0.53 |
| with PHC+ (2024) | 0.99 | 0.71 | 0.50 | 0.21 |
| open-loop | **1.0** | 0.8 | 0.19 | 0.23 |

Table 1: **Task success rates.** Bold and underscore relate to *multi-task* only. CLoSD significantly excels on *Striking* and *Get-up*, which require careful object interaction.

| | R-precision ↑ (top-3) | FID ↓ | Runtime ↓ [msec] | Speed ↑ [fps] |
|---|---|---|---|---|
| DiP (ours) | 0.78 | 0.28 | 11.4 | $3.5 \cdot 10^3$ |
| 5 Diff-steps | 0.76 | 0.32 | 6.1 | $6.6 \cdot 10^3$ |
| 20 Diff-steps | **0.8** | 0.28 | 23 | $1.7 \cdot 10^3$ |
| $N_p = 40$ | 0.74 | 0.7 | 13 | $3.1 \cdot 10^3$ |
| $N_g = 20$ | 0.78 | **0.26** | 11.4 | $1.7 \cdot 10^3$ |
| $N_g = 80$ | 0.74 | 1.18 | 16 | $5 \cdot 10^3$ |

Table 2: **DiP ablation study.** We use DiP with prefix length $N_p = 20$, generation length $N_g = 40$, and 10 diffusion steps. DiP performs surprisingly well even with as few as 5 diffusion steps.

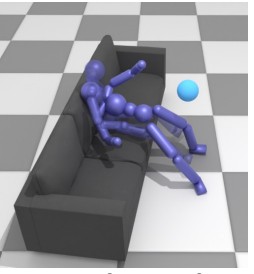 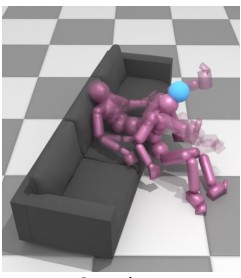 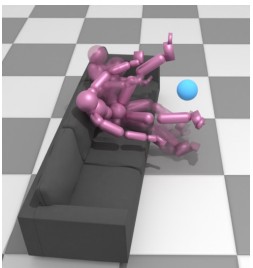 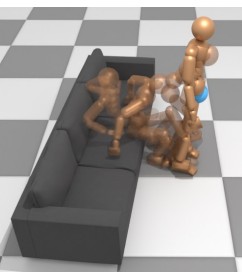

| UniHSI [Xiao 2024] | Open loop | Close loop (No fine-tuning) | Ours (with fine-tuning) |

Figure 4: **Comparisons of the getup task.** The pelvis target is marked in cyan. CLoSD is able to get up successfully with a human-like motion. Before fine-tuning on object interaction with the closed-loop, it was able to sit but struggled to get up. The open-loop baseline struggles with any interaction with the sofa due to the lack of re-planning. UniHSI (Xiao et al., 2024) was designed to minimize contact-point distance and thus lifts the pelvis instead of getting up from the sofa.

**Goal Reaching.** By specifying a random target for the *pelvis* joint on the XY plane (without forcing a specific height) while specifying a text prompt defining the desired locomotion type (e.g. running, walking, jumping, dancing waltz, etc.) and its style (e.g. "rapidly", "like a zombie", "slalom",), CLoSD performers goal-reaching. While the target can be far away, and the motion plan $x^{\text{pred}}$ is typically up to two seconds long, we heuristically limit the velocity of the character by assigning *intermediate targets* to the last frame of $x^{\text{pred}}$ for each DiP iteration. The success criterion of this task is when the pelvis joint XY location is at a distance of $0.3[m]$ from the target.

**Object Striking.** In this task, the character needs to strike down a standing kickboxing bag. The success criterion of this task is when the bag angle with the ground plane is less than $75°$. For each episode, we randomly sample one of the hands or feet end-effector joints and a text prompt aligned with it. For example, an optional text for the *left hand* might be "A person is punching the target with the left hand.". The target for the end-effector joint will be the center of the bag. For robustness, we randomly perturb the bag's location during both train and test.

**Sitting down.** In this task, the character needs to sit down on a sofa. The target for the pelvis joint is at the center of the sofa's seat and the text specifies sitting down. The success criterion of the task is when the pelvis joint is on top of the sitting area. For robustness, we randomly perturb the sofa's location and orientation during both train and test.

**Getting up.** In this task, the initial pose is sitting down on a sofa object and the goal is to get up to a standing position. The target for the pelvis is placed 0.3[m] in front of the sofa, at a standing height $(0.9[m])$. The text is designed appropriately, e.g. "A person is standing up from the sofa and remains standing.". The success criterion of this task is when the pelvis joint is at a distance of $0.3[m]$ from the target, which empirically implies successful standing up. In practice, we learn the *sitting down* and *getting up* on the same episode, when we start at a standing rest pose, a sitting target, and sitting text, then when the sitting success criterion is met, we wait for another two seconds and switch the target and text prompt to get up.

**Results.** Examples of behaviors learned by the models are presented in Figures 1, 2, and the web page video. Table 1 compares the success rate performance of CLoSD to single- and multi-task agents over 1K episodes per task. We trained UniHSI (Xiao et al., 2024) on our tasks for 12K PPO epochs until convergence. UniHSI models their multi-task agent by reaching a sequence of contact points. Without a semantic description of the task, UniHSI touches the target instead of striking it, and bending the pelvis instead of getting up, which leads to low success rates and is reflected in Figure 4. The open-loop baseline struggles to successfully perform the interaction tasks, particularly for the sitting and get-up tasks, due to the inability to adapt to the actual motion in progress. Using the motion tracking controller without fine-tuning (both our single-primitive version and PHC+ (Luo et al., 2024), as provided by its authors) for closed-loop behavior also results in significantly degraded performance for interaction tasks. To stress test the robustness of the learned sitting controls, we conducted an additional experiment in which we randomly perturbed the sit's height between 20 to $60[cm]$ at test time only. The sitting and getting up success rates dropped slightly, from $(0.88, 0.98)$ to $(0.83, 0.95)$ respectively.

| | Text-to-motion | | | | | | Physics-based metrics | | | |
| | R-precision ↑ | | | FID ↓ | Multi-Modal | Diversity | Physics | Penetration ↓ | Floating ↓ | Skating ↓ |
| | Top 1 | Top 2 | Top 3 | | Distance ↓ | | model | [mm] | [mm] | [mm] |
|---|---|---|---|---|---|---|---|---|---|---|
| Ground Truth | 0.405 | 0.632 | 0.746 | 0.001 | 2.95 | 9.51 | | 0.0 | 22.9 | $206 \cdot 10^{-3}$ |
| MDM (2023) | 0.406 | 0.603 | 0.719 | 0.423 | 3.53 | 9.52 | ✗ | 0.147 | 28.6 | $330 \cdot 10^{-3}$ |
| MoConVQ (2024) | 0.309 | 0.504 | 0.614 | 3.279 | 3.97 | 8.01 | ✓ | 0.249 | 32.0 | $294 \cdot 10^{-3}$ |
| CLoSD (Ours) | 0.385 | 0.578 | 0.693 | 1.755 | 3.65 | 8.23 | ✓ | 0.021 | 20.1 | $1.2 \cdot 10^{-3}$ |
| DiP only | 0.464 | 0.668 | 0.777 | 0.283 | 3.15 | 9.21 | ✗ | 0.083 | 23.6 | $629 \cdot 10^{-3}$ |
| shorter-loop | 0.303 | 0.473 | 0.586 | 3.481 | 4.18 | 7.93 | ✓ | 0.018 | 21.5 | $0.3 \cdot 10^{-3}$ |
| longer-loop | 0.369 | 0.559 | 0.674 | 1.671 | 3.72 | 8.20 | ✓ | 0.015 | 19.8 | $1.2 \cdot 10^{-3}$ |
| open-loop | 0.367 | 0.557 | 0.672 | 1.445 | 3.71 | 9.38 | ✓ | 0.007 | 19.9 | $1 \cdot 10^{-3}$ |
| w.o. fine-tuning | 0.367 | 0.555 | 0.670 | 2.154 | 3.75 | 9.34 | ✓ | 0.016 | 20.2 | $1.6 \cdot 10^{-3}$ |

Table 3: **Text-to-motion** on the HumanML3D benchmark (Guo et al., 2022), alongside the PhysDiff metrics (Yuan et al., 2023), which evaluate aspects of physical correctness. Diversity values closer to the ground truth are preferred.

## 5.3 TEXT-TO-MOTION EVALUATION

For text-to-motion evaluation, we disable the target conditioning and direct CLoSD to perform various behaviors by conditioning on text only. We use the popular HumanML3D benchmark, and compare our model to data-driven kinematic motion synthesis models and to MoConVQ Yao et al. (2024), which is a state-of-the-art physics-based text-to-motion controller.

**Text-to-motion metrics.** We evaluate the text-to-motion performance using the HumanML3D (Guo et al., 2022) benchmark, which is widely used for this task. The benchmark uses text and motion encoders, mapping to the same latent space and learned using contrastive loss. The following metrics are calculated in that latent space. *R-precision* measures the proximity of the motion to the text it was conditioned on. A generation is considered successful if its text is in the top-$n$ closest prompts compared to 31 randomly sampled negative examples. *FID* measures the distance of the generated motion distribution to the ground truth distribution, *Diversity* measures the variance of generated motion, and *MultiModel distance* is the mean $L2$ distance between the pairs of text and conditioned motion. For full details, we refer the reader to the original paper.

**Physics-based metrics.** We report three physics-based metrics suggested by Yuan et al. (2023) to evaluate the physical plausibility of generated motions: *Penetration* measures the average distance between the ground and the lowest body joint position below the ground. *Floating* measures the average distance between the ground and the lowest body joint position above the ground. We use a tolerance of 5mm to account for geometry approximation. *Skating* measures the average horizontal displacement within frames that demonstrate skating. Two adjacent frames are said to have skating if they both maintain contact to ground (minimum joint height ≤ 5mm).

**Results.** Table 3 compares CLoSD to current methods. Generally, physics-based text-to-motion significantly improves the physical metrics, but degrades the data distribution metrics. The latter is expected as data-driven models model the distribution directly. To evaluate MoConVQ (Yao et al., 2024), we collected samples using the code and model released by the authors, following their described procedures. In comparison to MoConVQ, which is a physics-based method, our method excels in all metrics. Our ablation study (bottom of Table 3) indicates that fine-tuning improves the results. Additionally, pure text-to-motion benefits from longer plans. Since closed-loop feedback is crucial for object interaction, we find our 2 second plans to be a good trade-off.

## 6 CONCLUSIONS AND LIMITATIONS

We have presented a method to combine the flexible planning capabilities of diffusion models with the grounded realism and responsiveness of physics-based simulations. CLoSD performs text-prompt directable multi-task sequences involving physical interactions with objects in the environment, including navigation, striking an object with specified hands or feet, and sitting down or getting up. Adding further scene awareness, i.e., vision or height maps, to the diffusion-based planner and the tracking controller will be important for achieving a wider range of tasks. Further investigation could tackle longer-time-scale planning, as we focused on mid- and low-level planning. Additionally, planning at a latent action space, such as the recent PULSE (Luo et al., 2024), instead of the pose space, might enable a more straightforward tracking. Lastly, further adaptivity could be incorporated into the loop. For example, a temporally adaptive loop could help with fast motions and settings where closer feedback is important. Conversely, longer tracking horizons could mitigate occasional artifacts.

ACKNOWLEDGEMENTS

This research was partly supported by the Israel Science Foundation (1337/22), Len Blavatnik and the Blavatnik family foundation, The Tel Aviv University Innovation Laboratories (TILabs), and the NSERC Discovery Grant (DGECR-2023-00280, RGPIN-2020-05929).

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
