# OpenReview forum: "CLoSD: Closing the Loop between Simulation and Diffusion for multi-task character control"
_ICLR.cc/2025/Conference — ICLR 2025 Spotlight_

### Official Review · Reviewer_N8zK · 2024-10-28

**Soundness:** 2
**Presentation:** 3
**Contribution:** 2
**Rating:** 6
**Confidence:** 4

**Summary:**

The paper presents CLoSD, a multi-task character motion generation approach that combines text-driven motion geneation models and reinforcement learning to control physics-based simulations of human-like motion. By combining motion diffusion, which generates a variety of text-based motions, and RL, which enables interaction with the environment, CLoSD ensures both diversity and physical accuracy in character motion. The system operates in a closed loop, where a Diffusion Planner (DiP) generates motion plans from text prompts and target locations, and a tracking controller uses these plans to adapt the character’s actions in real-time. This setup enables motion generation across tasks like navigating to a target, interacting with objects, and completing actions like sitting or getting up. The approach demonstrates high task success rates, adaptability to on-the-fly text changes, and a significant improvement over existing controllers, as it allows characters to perform varied tasks while adhering to realistic physical constraints.

**Strengths:**

1. The paper introduces a closed-loop integration between motion diffusion and RL, leveraging each method’s strengths and overcoming traditional limitations, such as the need for pre-planned motion sequences or manual fine-tuning.
2. CLoSD’s capability to execute multiple tasks through simple text prompts and target positions highlights its flexibility and generalization, reducing the need for task-specific controllers.
3. By optimizing the Diffusion Planner to be both autoregressive and fast, CLoSD maintains responsiveness in dynamic environments, a crucial factor for real-world applications.

**Weaknesses:**

1. The paper could benefit from better contextualization. While the integration of CAMDM and PHC for multitask character control is intriguing and effectively combines a diffusion model for language-conditioned motion generation with a tracker to ensure physical feasibility, the Related Work section would benefit from a deeper discussion since similar architecture has been explored in previous works like Trace and Pace or InsActor. Specifically, while it briefly mentions these methods, it lacks depth on the specific contributions of CLoSD that differ from such hierarchical decomposition in language-conditioned motion generation as explored in prior works. Including insights on how CLoSD advances or contrasts with these methods, such as motion in-painting, goal-reaching, and motion chaining demonstrated in InsActor, could enhance readers’ understanding of the unique contributions of this work.
2. The novelty of the work could be better conveyed. While the combination of CAMDM and PHC for multitask character control aligns with existing architectures, CLoSD’s main distinguishing feature is its real-time generation capability, which largely stems from the use of CAMDM. A clearer exposition of the unique contributions and advancements CLoSD introduces—beyond real-time generation—would help emphasize its originality and innovation.
3. The experimental evaluation is somewhat limited in scope. Although the paper includes experiments on motion generation with HumanML3D, a broader range of tasks would provide a more comprehensive validation of the approach. Since the method is intended to enable character interactions with environmental objects, including more diverse interaction tasks beyond the four basic ones presented would greatly enrich the empirical support and demonstrate the approach’s robustness and generalizability.

Overall, the paper presents an interesting approach to interactive character control. However, I would appreciate a deeper discussion that thoroughly compares the proposed work with prior works.

**Questions:**

See weaknesses

---

> ### Author Response · Authors · 2024-11-22
>
> Thank you for your thoughtful comments and questions.
>
> > 1. … Including insights on how CLoSD advances or contrasts with these methods, such as motion in-painting, goal-reaching, and motion chaining demonstrated in InsActor, could enhance readers’ understanding of the unique contributions of this work.
>
> Thank you for your suggestions! Compared to Trace & Pace, CLoSD's high diffusion plans generate whole-body humanoid motion while Trace & Pace generates 2D waypoints. Thus, Trace & Pace is limited to the trajectory following the task and does not have language control over the generated motion. Compared to InsActor, CLoSD runs in real-time and uses a shorter planning horizon, which means that the planner can react to feedback from the scene. Our planning design effectively closes the loop between the diffusion planner and the low-level tracker which enables the support of humanoid-object interaction. The differentiable simulator InsActor uses also leads to some artifacts, such as foot penetration and sliding, alleviated by CLoSD's choice of simulator and RL tracking policy. We have added these discussion points to the manuscript.
>
> > 2. The novelty of the work could be better conveyed. While the combination of CAMDM and PHC for multitask character control aligns with existing architectures, CLoSD’s main distinguishing feature is its real-time generation capability, which largely stems from the use of CAMDM. A clearer exposition of the unique contributions and advancements CLoSD introduces—beyond real-time generation—would help emphasize its originality and innovation.
>
> The paper's title, abstract, and introduction highlight its main contribution - closing the loop between kinematic motion generation and tracking policy achieves a robust multi-task agent, controlled by text. Using a real-time motion generator is thus critical and hence we follow the principles in CAMDM. Indeed CAMDM is a great work, nonetheless, this model is conditioned on style class and trajectory, which are insufficient to describe tasks such as those presented in this paper. Instead, DiP has text and target control which make it more expressive and better grounded to the task.
>
> > 3. … Since the method is intended to enable character interactions with environmental objects, including more diverse interaction tasks beyond the four basic ones presented would greatly enrich the empirical support and demonstrate the approach’s robustness and generalizability.
>
> As you point out, CLoSD was demonstrated on four popular RL tasks. A single CLoSD agent outperforms prior work, without requiring additional reward engineering. In addition, the technique and style of solving the tasks can be controlled intuitively using text; for example, striking is performed by kicking, pushing, or punching according to the text prompt. We believe that the general and robust design of CLoSD could be used for more tasks and leave this as future work.

---

> > ### Comment · Reviewer_N8zK · 2024-11-27
> >
> > Thank you to the authors for the detailed response! The modifications have made the work better contextualized, and I will maintain my current rating.

---

### Official Review · Reviewer_1o2G · 2024-11-01

**Soundness:** 3
**Presentation:** 3
**Contribution:** 3
**Rating:** 8
**Confidence:** 4

**Summary:**

This paper presents CloSD, a method that combines a diffusion planner with a reinforcement learning tracking policy to enable multi-task character control. The results demonstrate that CloSD achieves natural and versatile motions.

**Strengths:**

- This paper combines a diffusion planner with an RL motion tracker to achieve natural and versatile physics-based character control across multiple tasks.
- CloSD outperforms previous methods, such as InterPhys and UniHSI, and provide a solid framework for future research.
- Extensive results and ablation studies demonstrate the effectiveness of each design choices and influence of hyper-parameters.
- The paper is overall well-structured and well-written.

**Weaknesses:**

Please refer to the questions part.

**Questions:**

- I am curious about the difference between open-loop and closed-loop approaches. In Table 1, the open-loop setting performs well on “goal reaching” and “object striking” tasks but struggles with “sitting” and “getting up” tasks. Could you elaborate on why re-planning is necessary in these cases?
- $x^{\text {prefix }}$ is defined as “current frames,” but does this risk the model forgetting its previous plans? In other words, might it be more effective to define $x^{\text {prefix }}$ as motion tracking errors, or are there alternative design choices that could work better?

---

> ### Author Response · Authors · 2024-11-22
>
> Thank you for your thoughtful comments and questions.
>
> > difference between open-loop and closed-loop approaches… the open-loop setting performs well on “goal reaching” and “object striking” tasks but struggles with “sitting” and “getting up” tasks. Could you elaborate on why re-planning is necessary?
>
> As you pointed out, the open-loop configuration can perform reaching and striking but fails to sit and get up. For reaching and striking, the object interaction is minimal, and the controller is robust enough to cover for DiP inaccuracies. For the sofa interactions, re-planning is essential. The open loop plan quickly accumulates errors that the controller cannot recover from. The ablations at the end of the supplementary video demonstrate this failure case.
>
> > $x_{prefix}$  is defined as “current frames,” but does this risk the model forgetting its previous plans? In other words, might it be more effective to define $x_{prefix}$ as motion tracking errors, or are there alternative design choices that could work better?
>
> The main reason for feeding DiP with the actual performed frames is to let it re-plan, i.e. tune the plan according to the motion performed in practice. For some complex long-horizon tasks, feeding DiP with the previously predicted frames and the frames performed by the tracking controller might be beneficial. Following your comment, we extended this discussion in the conclusions section in blue.

---

### Official Review · Reviewer_U9rt · 2024-11-02

**Soundness:** 3
**Presentation:** 3
**Contribution:** 3
**Rating:** 8
**Confidence:** 4

**Summary:**

This work presents a method that combines a diffusion-based motion planner with a tracking controller in a closed-loop setup. The motion planner is an autoregressive diffusion model that predicts future character poses based on previous states and conditioning inputs, such as joint target positions and text descriptions. The controller is an off-the-shelf state-of-the-art tracking controller finetuned to the motions generated by the DiP. The result is a text-to-motion controller that can be used to solve different tasks while having control over the style of the motions.

**Strengths:**

* The goal of the paper is clear: combine the strength of a DIffusion Model (MDM) with a physics-based controller (PHC).
* The method is simple yet interesting enough.
* The paper is well-written, and the method is clearly explained.
* Code will be publicly available.

**Weaknesses:**

* While the method is well explained, the experimental section lacks clarity and precision. (see *Questions*)
* The experiments could be better designed.
* The video shows lower motion quality than state-of-the-art physics-based character control, which the paper does not address.
* The limitations are not clearly stated and discussed.

**Questions:**

**Tracking Policy**

I have a few questions regarding the PHC setup. It is stated that a simplified version of PHC was used, excluding the mixture of experts.
* Does this imply that only a single primitive policy was trained? If so, is the controller essentially a standard policy trained solely with AMP and tracking rewards?
* Additionally, how would the full PHC version perform in comparison?
* Would an alternative controller, such as the one proposed by Chentanez et al. 2018, which incorporates future frames, potentially be more effective?

There has been more recent work on character control for tracking reference motions, including newer controllers developed in 2024.
* Consider referencing newer motion tracking controllers to provide a more current perspective.
* Additionally, other methods have also explored controllers that operate without a reference motion. While these are not directly relevant, It would be beneficial to discuss such approaches, including whether the integration of a DiP-like approach is feasible in this context.

**Comparisons with UniHSI**

UniHSI uses an LLM with scene information to design a chain of contacts. Their demonstrated results look less smooth than CLoSD; however, they look much better than those presented here. For example, their pipeline shows successful „stand-up “ performance. As far as I understand, in the comparisons here, the LLM Planner was omitted, and instead, a fixed target label for the pelvis alone is defined.
* The discrepancy between UniHSI’s original results and the lower-quality outcomes here needs clearer explanation and precise clarification.
* Further UniHSI shows more complex chains of tasks, sitting on different objects, lying down etc. Can CLoSD also sit on variable hights? where are the limits of versatility?

**Text-to-Motion**
* PhysDiff’s metrics are used to evaluate the physical correctness of the text-to-motion quality compared to MDM and MoConVQ. It seems more intuitive to use PhysDiff (MDM + Physical Projection) as an additional comparison to understand how much the closed-loop approach improves over the proposed physical projection in PhysDiff.
* As far as I understand, the CLoSD results are the motions performed in the physics-based simulator. This makes me wonder why there is still such a high floating error of 2cm on average.
* It would be helpful to visualize a side-by-side comparison of DiP and MDM motions, as DiP appears to significantly outperform MDM in this aspect.
* A comparison between DiP and CAMDM is needed to clarify why DiP might be better suited than the Chen et al. 2024 architecture mentioned in line 236.

**Motion Quality**
* The presented motions in the video are not very smooth and contain a lot of shakiness and artifacts that seem to trick the physics (e.g., fast vibrations allowing the character to drift - 02:42)
* In the renderings with the SMPL mesh, strong artifacts in hand pose and feet are visible. Can you explain what is causing those artifacts that are less severe in MDM.
* In the last motion in the video, “A person is waving goodbye,” the model attempts a simple motion that MDM typically handles with high precision, yet here it appears less accurate. This raises the question of whether CLoSD is overfitted to the specific tasks, potentially limiting its ability to leverage the full capabilities of DiP/MDM in terms of motion quality and diversity.

**Limitations**
* MDM can generate a diverse range of dynamic motions, and PHC can track many AMASS motions. However, the motions presented here appear more constrained. A more detailed explanation of the method’s limitations would be helpful.
* The method’s applications rely heavily on various tricks, making it challenging to determine how much of the outcome is due to human engineering versus the method’s inherent capabilities. For instance, to ensure the object is hit, the user must specify a joint target location and an appropriate prompt. Consequently, the approach appears overfitted to this specific set of motions/tasks and does not fully utilize the generative capabilities of the diffusion model. It would be helpful if the authors could discuss these limitations further.

---

> ### Author Response · Authors · 2024-11-22
> **PART 1/3**
>
> Thank you for your thoughtful comments and questions.
>
> ### Tracking Policy
>
> > …Does this imply that only a single primitive policy was trained? If so, is the controller essentially a standard policy trained solely with AMP and tracking rewards?
>
> Indeed we use a single primitive PHC. Its authors find, retrospectively, that it is effective (https://github.com/ZhengyiLuo/PHC?tab=readme-ov-file#current-results-on-cleaned-amass-11313-sequences) in terms of motion tracking if trained for longer, while the mixture-of-expert helps provide failure-state recovery capability. The single primitive is a motion tracking policy, trained with the imitation reward and AMP style reward. Hard negative mining similar to the one proposed in PHC+ is used to train our tracker. We chose the single-primitive tracker as it is easier to fine-tune for human-object interaction tasks.
>
> > Additionally, how would the full PHC version perform in comparison?
>
> Following your suggestion, we evaluated CLoSD using the original multi-primitive PHC+ as released by the authors and added the results to Table 1. We observe that it performs slightly worse than the ‘w.o. Finetuning’ experiment with our single-primitive version.
>
> > Would an alternative controller, such as the one proposed by Chentanez et al. 2018, which incorporates future frames, potentially be more effective?
>
> Excellent suggestion. Trackers that incorporate more future frames, like the one proposed by Chentanez et al. 2018, should provide the tracking policy with more information to prepare for dynamic motions. However, as noticed in UniCon [Wang et al. 2020], feeding too many future frames could slow down network training as too much information is being fed to the RL policy. A few future frames (e.g., one frame used in PHC) could lead to comparable results. How to effectively use future frames in an RL framework is an important revenue for future exploration.
>
> > Consider referencing newer motion tracking controllers to provide a more current perspective.
>
> There have been new neural motion controllers like MaskedMimic [Tessler et al. 2024], which we plan to leverage for its built-in terrain awareness; thanks for the suggestion! Please let us know if there is any other specific work you have in mind!
>
> > Additionally, other methods have also explored controllers that operate without a reference motion. While these are not directly relevant, It would be beneficial to discuss such approaches, including whether the integration of a DiP-like approach is feasible in this context.
>
> As pointed out, works like PULSE [Luo et al. 2024] provide a latent action space that might be beneficial for future versions of DiP; we added this point to the conclusions section.
>
> ### Comparisons with UniHSI
>
> > As far as I understand, in the comparisons here, the LLM Planner was omitted, and instead, a fixed target label for the pelvis alone is defined.
>
> For the UniHSI experiment,  we used the LLM plans published by the authors that were most similar to our tasks and manually modified them to fit our objects.
>
> >  For example, their pipeline shows successful „stand-up “ performance.
>
> UniHSI demonstrated one standing-up motion but did not statistically evaluate this task (See Table 3 in their paper). Our evaluation shows that UniHSI succeeds at getting up with 8% success (see Table 1). In the video comparison, we show the UniHSI common failure case, which bends at the pelvis without getting up, which is a result of local minima of their goal-reaching objective.
>
> > The discrepancy between UniHSI’s original results and the lower-quality outcomes here needs clearer explanation and precise clarification.
>
> We trained UniHSI using their public code, assisted by the authors. Our and their quantitative evaluation share two tasks - sitting and reaching. In both their and our evaluation setups (Table 1 [in our paper] and Table 3 [in their paper]), those tasks achieve similar success rates, which validates the UniHSI adaptation presented in our paper. We further show that UniHSI fails in the striking and getting up tasks, which were not quantitatively evaluated in the original paper.
>
> > Can CLoSD also sit on variable heights? where are the limits of versatility?
>
> Regarding variable sitting heights - we agree and hence add a variable height experiment (see in blue in section 5.2). The experiment indicates that CLoSD is robust to different heights.

---

> ### Author Response · Authors · 2024-11-22
> **PART 2/3**
>
> ### Text-to-Motion
>
> > PhysDiff’s metrics are used to evaluate the physical correctness of the text-to-motion quality compared to MDM and MoConVQ. It seems more intuitive to use PhysDiff (MDM + Physical Projection) as an additional comparison to understand how much the closed-loop approach improves over the proposed physical projection in PhysDiff.
>
> We agree that a comparison with PhysDiff would be insightful. However, since the authors did not publish their code, we could not reproduce the experiments despite our best efforts. Nonetheless, with the guidance of the PhysDiff authors, we were able to re-implement their metrics and will make them available with our code.  Notice that PhysDiff utilizes residual force control (applying a virtual force on the character's root for stabilization).
>
> > As far as I understand, the CLoSD results are the motions performed in the physics-based simulator. This makes me wonder why there is still such a high floating error of 2cm on average.
>
> We measure ‘floating’, according to PhysDiff, as the average distance between the ground and the lowest body joint position above the ground. As a result, natural motions with intentional jumping and running, for example, will result in some value for floating. This phenomenon is reflected in the floating value of 22.9 mm for ground-truth data. An alternate interpretation for the `floating` would be “closer-to-gt-is-better” rather than “lower-is-better”. And as can be seen from Table.3, CLoSD is producing the closest-to-ground-truth value for floating compared to prior work.
>
> > It would be helpful to visualize a side-by-side comparison of DiP and MDM motions, as DiP appears to significantly outperform MDM in this aspect.
>
> We agree and will add MDM and DiP side-by-side visualizations in our next revision. Admittedly, although DiP has slightly better FID and R-precision, visually their results are comparable.
>
> > A comparison between DiP and CAMDM is needed to clarify why DiP might be better suited than the Chen et al. 2024 architecture mentioned in line 236.
>
> The main difference between DiP and CAMDM is the control signals. CAMDM is conditioned on style class and trajectory, which are insufficient to describe tasks such as those presented in this paper. Instead, DiP has text and target control which make it more expressive and better grounded to the task. As a result, CAMDM cannot be directly integrated into the CLoSD framework. If you have an idea of how to perform such a comparison please share it with us and we will try to add it to our next revision.
>
> ### Motion Quality
>
> > The presented motions in the video are not very smooth and contain a lot of shakiness and artifacts that seem to trick the physics
>
> We agree that CLoSD has some observable jitters and discuss how this might be resolved in future work in the conclusions and limitations section.
>
> > In the renderings with the SMPL mesh, strong artifacts in hand pose and feet are visible. Can you explain what is causing those artifacts that are less severe in MDM.
>
> Thank you for spotting those SMPL artifacts. Our simulated SMPL humanoid may utilize the toe joint to stabilize during movements, and such motion is not properly handled when converting it to mesh.  We will fix that and re-render the SMPL mesh in our next revision.
>
> > In the last motion in the video, “A person is waving goodbye,” the model attempts a simple motion that MDM typically handles with high precision, yet here it appears less accurate. This raises the question of whether CLoSD is overfitted to the specific tasks, potentially limiting its ability to leverage the full capabilities of DiP/MDM in terms of motion quality and diversity.
>
> Indeed, CLoSD can generate better “waving goodbye” motions than the one presented. It was chosen in a rush and will be replaced. Not all MDM generations are accurate, either. In any case, this is a just-for-fun motion, and the paper includes a much more comprehensive evaluation of the text-to-motion abilities of CLoSD. First, we let CLoSD perform all the HumanML3D test set and evaluate it in Table 3. As expected, CLoSD presents slightly degraded FID and R-precision compared to DiP-only as it is more constrained, yet significantly improves the physics-based metrics. Moreover, it outperforms MoConVQ in both data-driven and physics-based metrics. In addition, our video demonstrates multiple text conditions for each task, including various locomotions for the reaching task.

---

> ### Author Response · Authors · 2024-11-22
> **PART 3/3**
>
> ### Limitations
>
> > MDM can generate a diverse range of dynamic motions, and PHC can track many AMASS motions. However, the motions presented here appear more constrained. A more detailed explanation of the method’s limitations would be helpful.
>
> We evaluate CLoSD text-to-motion abilities and task success rate performance and find it outperforms prior methods. Yet, as you pointed out, there are limitations regarding motion quality. Following your comment, we extended the discussion of limitations in the conclusions section in blue.
>
> > The method’s applications rely heavily on various tricks, making it challenging to determine how much of the outcome is due to human engineering versus the method’s inherent capabilities. For instance, to ensure the object is hit, the user must specify a joint target location and an appropriate prompt. Consequently, the approach appears overfitted to this specific set of motions/tasks and does not fully utilize the generative capabilities of the diffusion model. It would be helpful if the authors could discuss these limitations further.
>
> Regarding the striking task, the user specifies the striking method using text, and the striking location is automatically determined according to the object location, which is included in the simulator state. We view the text as the control signal of CLoSD as much as MVAE, ASE, and others are controlled using arrow keys. Regretfully, we disagree that CLoSD overfits. It presents good text-to-motion performance while solving four different tasks with a high success rate. We agree that CLoSD can be more versatile in the future, for example, using a richer observation space and more precise control. Following your comment, we extend this discussion in the conclusions section (in blue).

---

> ### Comment · Reviewer_U9rt · 2024-11-24
>
> I appreciate the authors' thoughtful response and their effort in addressing the concerns. The additional experiments and clarifications have significantly enhanced the quality of the paper. I have no further questions and have accordingly increased my score.

---

> > ### Author Response · Authors · 2024-11-25
> >
> > Thank you U9rt, this is much appreciated!

---

### Official Review · Reviewer_MtoF · 2024-11-02

**Soundness:** 3
**Presentation:** 3
**Contribution:** 3
**Rating:** 8
**Confidence:** 3

**Summary:**

This paper presents CLoSD, a text-driven reinforcement learning (RL) controller for character motion, merging the generative capabilities of diffusion models with the physical realism of RL-based control in a closed-loop framework. The proposed framework has two components: a Diffuion Planner (DiP), and a RL tracking controller. DiP is an auto-regressive diffusion model that is controlled by textual prompts and target locations and outputs motion plans. The RL tracking controller, based on the Perpetual Humanoid Control (PHC) model, interprets DiP’s motion plans and executes them in a physics simulation.

**Strengths:**

The paper is well-written, with the key contributions listed clearly in the introduction. The authors also identified relevant literature in this line of research and highlighted their contributions in the method. The authors also included low-level technical details which aid to the reproducibility of the work. Experimental evaluations highlight CLoSD’s superior performance on multi-task benchmarks, including goal-reaching, object striking, sitting and getting up. In tasks requiring environmental interaction, CLoSD significantly outperforms both state-of-the-art open-loop diffusion models and other text-to-motion RL systems, demonstrating its ability to maintain stable, responsive control across diverse physical interactions. Overall, I believe this paper is a good step towards understanding the interaction between generative models and physical simulation.

**Weaknesses:**

Recent generative models are capable of generating plans in much larger observation spaces. Can the authors comment on how this work can involve sensory inputs (if at all), such as vision, to enhance CLoSD's adaptability in more complex environments? Moreover, can the authors comment on the potential failure modes of their method if their method was used for a longer planning horizon to generate even more intricate motion sequences?

**Questions:**

Please see weaknesses above.

---

> ### Author Response · Authors · 2024-11-22
>
> Thank you for your thoughtful comments and questions.
>
> > Larger observation spaces… how this work can involve sensory inputs (if at all), such as vision, to enhance CLoSD…?
>
> Good point. We agree that enriching the observation space is necessary to support more interactions. This will require the controller and the planner to be informed of those observations. DiP is a first step toward that goal, as the task’s target location informs DiP with the objectives. We believe that integrating scene-aware motion diffusion models, such as the recent TeSMo [Yi et al. 2024] will benefit CLoSD in the future. See more examples in the related work section. Following your comment, we extended this discussion in the conclusions section in blue.
>
> > Potential failure modes of their method… for a longer planning horizon to generate even more intricate motion sequences?
>
> The length of the planning horizon draws a trade-off between the quality of tracking and planning. For example, limiting the planning horizon to 0.25 seconds empirically prevents CLoSD from kicking properly, as this action requires at least one second of planning ahead. We note this trade-off in our experiments and conclusions section. As you correctly note, some tasks require an even longer planning horizon. In this work, we demonstrated high-level planning with a simple state machine as a first step. We believe that developing a more comprehensive high-level planner will be an important step toward more complex tasks. Following your comment, we extended this discussion in the conclusions section in blue.

---

> > ### Comment · Reviewer_MtoF · 2024-12-03
> > **Response to Authors**
> >
> > I thank the authors for their response. I will keep my current score recommending acceptance for this paper.

---

### Author Response · Authors · 2024-11-22
**General Response**

We thank the reviewers for their comprehensive reviews and positive feedback.
Thank you for noting our work “achieve natural and versatile physics-based character control across multiple tasks” (1o2G), “overcoming traditional limitations, such as the need for pre-planned motion sequences or manual fine-tuning” (N8zK), “outperforms previous methods” (1o2G), and “demonstrates high task success rates, adaptability to on-the-fly text changes, and a significant improvement over existing controllers” (N8zK). MtoF concluded that “this paper is a good step towards understanding the interaction between generative models and physical simulation.”. You also mentioned that the paper is well-written (MtoF, U9rt, 1o2G), and the method is clearly explained (MtoF, U9rt).

We have revised the paper according to your feedback (changes are highlighted in blue) and provided more detailed responses in the individual messages.

---

### Meta-Review · Area_Chair_Ph7C · 2024-12-16

**Metareview:**

The paper presents CLoSD,  method combining motion diffusion models with RL-based control for physics-based character motion generation. The key contribution is a closed-loop framework integrating a Diffusion Planner (DiP) and tracking controller for real-time text-driven motion generation.

Strengths:
+ Novel integration of diffusion models and RL to achieve fast, closed-loop simulated character motion generation.
+ Strong experimental results on diverse scenarios and motion complexity
+ Well-documented implementation details

Weaknesses:
- Initial concerns about comparison with and differentiation from prior work like Trace & Pace and InsActor, though addressed in discussion
- Motion artifacts and jitters in some cases, particularly in SMPL mesh visualization
- Limited experimental evaluation scope, though expanded through additional tests
- Some technical details needed clarification, especially around control signal choices

The reviewers recommended acceptance after authors provided thorough responses addressing their concerns.

**Additional Comments On Reviewer Discussion:**

Initial Main Comments:
- Reviewer MtoF (Final Rating: 8) raised questions about sensory inputs and longer planning horizons
- Reviewer U9rt (Final Rating: 8) requested clarification on experiments, motion quality and UniHSI comparison
- Reviewer 1o2G (Final Rating: 8) asked about open vs closed-loop differences and design choices
- Reviewer N8zK (Final Rating: 6) questioned novelty relative to prior work and task scope

Key changes from rebuttal include: extended discussion comparing with prior work especially around real-time vs offline planning, clarification of design choices like planning horizon and control signals, explanation of motion quality artifacts, and demonstration of variable height capabilities. The authors effectively addressed concerns about novelty by emphasizing their unique closed-loop integration and real-time capabilities.

All reviewers were satisfied with the responses and maintained their recommendations for acceptance. U9rt specifically noted the responses "significantly enhanced the quality of the paper."

---

### Decision · Program_Chairs · 2025-01-22

Accept (Spotlight)